# Energy-Efficient Cooperative Transmission in Ultra-Dense Millimeter-Wave Network: Multi-Agent Q-Learning Approach

**DOI:** 10.3390/s24237750

**Published:** 2024-12-04

**Authors:** Seung-Yeon Kim, Haneul Ko

**Affiliations:** 1Department of Computer Convergence Software, Korea University, Sejong 30019, Republic of Korea; kimsy8011@korea.ac.kr; 2Department of Electronic Engineering, Kyung Hee University, Yongin-si 17104, Republic of Korea

**Keywords:** millimeter wave, cooperative transmission, B5G, power control, multi-agent Q-learning

## Abstract

In beyond fifth-generation networks, millimeter wave (mmWave) is considered a promising technology that can offer high data rates. However, due to inter-cell interference at cell boundaries, it is difficult to achieve a high signal-to-interference-plus-noise ratio (SINR) among users in an ultra-dense mmWave network environment (UDmN). In this paper, we solve this problem with the cooperative transmission technique to provide high SINR to users. Using coordinated multi-point transmission (CoMP) with the joint transmission (JT) strategy as a cooperation diversity technique can provide users with higher data rates through multiple desired signals. Nonetheless, cooperative transmissions between multiple base stations (BSs) lead to increased energy consumption. Therefore, we propose a multi-agent Q-learning-based power control scheme in UDmN. To satisfy the quality of service (QoS) requirements of users and decrease the energy consumption of networks, we define a reward function while considering the outage and energy efficiency of each BS. The results show that our scheme can achieve optimal transmission power and significantly improved network energy efficiency compared with conventional algorithms such as no transmit power control and random control. Additionally, we validate that leveraging channel state information to determine the participation of each BS in power control contributes to enhanced overall performance.

## 1. Introduction

In beyond fifth-generation (B5G) networks, millimeter wave (mmWave) technology is expected to be crucial due to its ability to support exceptionally high data rates and its vast spectrum availability [1]. The integration of mmWave with conventional low-frequency communications can offer both high performance and reliability, necessitating significant architectural and protocol adaptations across different network layers [2]. It is crucial to take this sort of holistic approach to ensure continuous and reliable connectivity in dense and highly mobile environments, a critical requirement for emerging applications such as Industry 4.0, vehicle-to-everything (V2X), and augmented reality (AR) [3]. Further, in mmWave networks, densely deploying multiple small base stations (mSBSs) can significantly enhance network capacity [4]. However, in ultra-dense mmWave networks (UDmNs), the dense proximity of mSBSs leads to overlapping coverage areas where signals from neighboring cells can interfere with each other, potentially degrading overall network performance [5]. The effective management of intercell interference (ICI) is crucial for maintaining the high data rates and reliability promised by UDmN [6].

In this context, Coordinated Multi-Point with Joint Transmission (CoMP-JT) in UDmN involves multiple BSs working together to serve users, particularly at cell edges where ICI is the most problematic [7]. By coordinating their transmissions, BSs can turn interference into desired signals, where the user coherently receives the desired signal from not only its serving BS but also an adjacent BS [8]. This approach is considered particularly valuable in densely deployed networks, where the potential for interference is high due to the proximity of the BSs [9].

Although CoMP-JT in UDmN can provide high data rates, it requires efficient energy management due to the additional cooperative transmissions between multiple mSBSs [10]. Therefore, in this paper, we propose a power control scheme that uses a cooperative transmission technique to enhance the energy efficiency of UDmN.

## 2. Related Work and Main Contribution

Several previous studies have attempted to improve the energy efficiency of mSBS with cooperative transmission in ultra-dense networks (UDNs). For example, refs. [11,12,13,14,15] have shown that the cooperative transmission-based mmWave network architecture can significantly improve not only the signal to noise and interference ratio (SINR) for the target user but also the energy efficiency of mSBSs. In [11,12], the authors proposed a user-centric cloud-based mmWave network in ultra-dense deployment scenarios, where each user can receive service from different BSs depending on their location. They also provided an analytical model based on the cooperative transmission. Kim et al. [13] showed the impact of energy saving for network architectures with multiple BSs using CoMP. Meanwhile, the authors in [14,15] proposed a power allocation scheme in a CoMP-enabled mmWave network, where BSs can use both renewable power and grid power to maximize the system energy efficiency. In [14], BSs can perform cooperative transmission when they have sufficient energy levels. In [15], mSBSs selectively perform cooperative transmission based on the behavior of neighboring mSBSs to minimize grid power.

Additionally, within a dense network, several algorithms have been proposed in [16,17,18,19,20] to obtain policies that maximize the energy efficiency. To elaborate, Liu et al. [16] proposed an energy-efficient cooperative transmission scheme in the mmWave network, where a user can select the joint transmission reception point (TRP) based on imperfect channel state information and delay constraints. In that scheme, the Kuhn–Munkres algorithm is used to find TRP. The authors in [17] introduced an optimization model for CoMP transmission in a dense mmWave network, where user association and power allocation are considered, and Lagrangian dual decomposition is adopted to solve the optimization model. Meanwhile, Sana et al. [18] proposed a distributed solution based on multi-agent reinforcement learning (RL), where UEs learn by experience to maximize the reward function with network sum rate. Then, Ju et al. [19] introduced an energy-efficient BS selection scheme in UDN, where each BS determines its operation mode, such as active or sleep, based on a policy obtained from deep RL, with the aim of maximizing the cumulative reward while minimizing the total power consumption of the network. Lastly, Iqbal et al. [20] proposed a cooperative Q-Learning (QL) algorithm for efficient joint radio resource management in UDN to handle interferences by adaptive power allocation while considering the minimum quality of service requirements. The optimal transmission power of mSBSs using the cooperative transmission technique is one of the most significant problems hindering performance enhancements in the energy efficiency of UDmN.

The multi-agent QL approach has been applied to energy-efficient solutions [21,22,23]. For example, Lim et al. [21] proposed a QL-based cooperative algorithm to maximize network performance, such as sum rate and individual rate in multicell networks, where each agent learns and collaborates to identify the optimal BS positions for deployment. This method is applicable in scenarios with limited link capacity and power at the BS. Lee et al. [22] presented a collaborative QL approach for managing multiple UAVs in a wireless network. To minimize energy consumption, this approach considers dynamic user demand, interference among UAVs, and maintaining network coverage. In [23], the authors proposed a multi-agent deep QL approach to maximize energy efficiency performance by considering both grid power and renewable energy. This method enables IoT devices to make offloading decisions in time-varying environments.

However, no previous work has optimized the energy efficiency and outage probability for UDmN using CoMP-JT in a distributed manner. Although CoMP technology can provide not only the improved SINR but also the network requisite capacity, it may lead to the increased energy consumption of mSBSs. As an example, consider a situation where several mSBSs with the same transmission power conducting CoMP-JT to the user. In this situation, if it is assumed that the user has a sufficient SINR, then even if some mSBSs transmit signals with lower transmission power, the energy efficiency of these mSBSs will be decreased. Therefore, in our scheme, the cooperative mSBSs conduct transmission power control in a distributed manner that considers the energy consumption and user outages of mSBSs. To optimize the trade-off between energy consumption and user outages, we use a distributed RL approach in UDmN. Moreover, to implement an algorithm with low complexity, a multi-agent tabular QL approach is adopted. The results demonstrate that our scheme can achieve optimal transmission power and significantly improve network energy efficiency compared to conventional algorithms, such as those without transmit power control or with random control.

The main contributions of our paper can be summarized as follows: (1) we propose an energy-efficient cooperative transmission power control scheme in UDmN using CoMP-JT, in which the cooperative mSBSs conduct the transmission power control in a distributed manner; (2) unlike conventional centralized approaches, we develop a policy that employs a multi-agent tabular QL-based cooperative transmission power control strategy; (3) through intensive simulations, the proposed scheme is demonstrated to outperform benchmarks in various environments. The rest of the paper is organized as follows: Section 3 describes the UDmN using CoMP-JT. Section 4 presents the formulation of the multi-agent QL model. Section 5 provides numerical examples, and concluding remarks are provided in Section 6.

## 3. System Model

Figure 1 shows the scenario of power control for UDmN, wherein the user can receive the desired multiple signals from cooperative mSBSs (e.g., mSBS_1_, mSBS_2_, mSBS_3_, and mSBS_4_). For our scenario, we consider mSBSs with varying transmission power levels, where each mSBS should control its own transmission power level to provide the user with sufficient SINR and minimize the energy consumption in UDmN.

### 3.1. SINR Model of User

We consider a propagation model consisting of a path loss component and a small-scale fading component, wherein Nakagami fading is applied to mmWave links [15]. The received signal power at a user from an mSBS *i*, i.e., mSBS_*i*_, is expressed as
(1)$$\begin{array}{c} P_{i}=P_{t}r_{i}^{-\varsigma }h_{i}, \end{array}$$
where $r_{i}^{-\varsigma }$ denotes path loss with path loss exponent ς at a distance $r_{i}$ between mSBS *i* and a user. $h_{i}$, as small-scale fading, is a normalized Gamma random variable with parameter *m*, i.e., $h_{i}∼\Gamma \left(m,1/m\right)$. Furthermore, $P_{t}$ is the transmit power of mSBS, which is divided into discrete powers as follows:(2)$$\begin{array}{c} P_{t}=\left\{p_{1},p_{2},\ldots ,p_{L}\right\}, \end{array}$$
where $p_{1}<p_{2}<\cdots <p_{L}$, and *L* represents the number of the power level. In our scheme, the transmit power of the mSBS is adjusted by its own power control policy.

For *n* neighboring mSBSs that can conduct cooperative transmission, based on Equation (1), the SINR of the average power received at a user is as follows:(3)$$\begin{array}{c} SINR=\frac{\sum_{i=1}^{n}P_{i}}{\sum I+\sigma^{2}}, \end{array}$$
where ∑I and $\sigma^{2}$ respectively denote the sum of the interference power and the additive noise.

### 3.2. Outage and Power Efficiency Gain Model

To evaluate the system performance, we consider the outage and power efficiency of mSBS. An outage occurs when the SINR of the user is under threshold $\gamma_{th}$. The power efficiency gain, $P_{g}$, for an mSBS, defined as the ratio of the deference between the maximum transmission power and controlled transmission power to the maximum transmission power, can be calculated as follows:(4)$$\begin{array}{c} P_{g}=\frac{\left(p_{L}-p_{k}\right)}{p_{L}}, \end{array}$$
where $p_{L}$ and $p_{k}$ are the maximum transmission power and *k*th transmission power of mSBS, respectively.

## 4. Multi-Agent Q-Learning Framework

The power control strategy of cooperative transmission can be modeled as a Markov decision process (MDP) and solved using an RL approach [21,24]. Note that in our system, each mSBS selects an action based on the current state, and the environment then transitions to the next state. The next state only depends on the action and the current state and is not related to previous states and actions. In our system, a centralized RL algorithm requires a central controller with complete information about multiple mSBSs, which leads to increased algorithmic complexity as the number of cooperative mSBSs grows. Extra connections between the central controller and the mSBSs are required to collect information on the BSs. To overcome these limitations of the centralized approach, we propose a multi-agent distributed QL approach to individually control the transmission power of the cooperative mSBSs.

For the proposed multi-agent QL framework as shown in Figure 1, agents, states, actions, and rewards are defined as follows:

**Agent:** Each cooperative mSBS is considered an agent in the proposed multi-agent RL framework. In an ever-changing environment, the agent takes action $a^{\left(t\right)}\in A$ in consideration of its current state $s^{\left(t\right)}\in S$ at each iteration *t*, and then obtains the corresponding reward $R^{\left(t\right)}$ and moves into the next state $s^{\left(t+1\right)}$.

**State (S):** We define the transmission power levels of mSBS as the state $s^{\left(t\right)}$ of the proposed framework. From Equation (2), the state of mSBS *i* can be represented as
(5)$$\begin{array}{c} s^{\left(t\right)}=p_{k},p_{k}\in p_{t}. \end{array}$$

**Action (A):** As mentioned in Section 2, each mSBS can control the transmission power within the entire state set represented by the transmission power levels. In our system, each agent has *L* options when taking an action, i.e., $a^{\left(t\right)}=p_{k},p_{k}\in p_{t}$. Moreover, to obtain the optimal policy for the power control, we utilize a decayed-epsilon greedy policy in which an action is randomly selected with a probability of ϵ. ϵ can be obtained as
(6)$$\begin{array}{c} \epsilon =\epsilon_{0}{\left(1-\epsilon_{0}\right)}^{\frac{e_{i}}{\xi N_{a}}}, \end{array}$$
where $\epsilon_{0}$ is the initial value of ϵ, $e_{i}$ is the current episode index, ξ is the exploration parameter, and $N_{a}$ is the total number of actions.

**Reward (R):** The reward is related to whether the outage and power efficiency gain are satisfied by dynamically adjusting the transmission power level of each mSBS for given interferences. Therefore, the reward function at each iteration *t* is defined as the weighted sum of the outage probability, $P_{out}^{\left(t\right)}$, and the power efficiency gain, $P_{g}^{\left(t\right)}$. It can be evaluated by
(7)$$\begin{array}{c} R^{\left(t\right)}=\beta \left(1-P_{out}^{\left(t\right)}\right)+\left(1-\beta \right)P_{g}^{\left(t\right)}, \end{array}$$
where β is the importance weight.

**Q-table (Q):** A Q-table, $Q^{\left(t\right)}$, reflects the value of the reward when the agent takes an action in each state at each iteration *t*. It represents a policy of which action the agent should choose in a given state. For QL, $Q^{\left(t\right)}$ is calculated using the following iterative procedure:(8)$$\begin{array}{cc} Q^{\left(t+1\right)}\left(s^{\left(t\right)},a^{\left(t\right)}\right) & =\left(1-\alpha \right)Q^{\left(t\right)}\left(s^{\left(t\right)},a^{\left(t\right)}\right)\\ \; & +\alpha \left[R+\gamma \max_{a^{\left(t+1\right)}\in A}Q^{\left(t\right)}\left(s^{\left(t+1\right)},a^{\left(t+1\right)}\right)\right], \end{array}$$
where α is the learning rate and γ is the discount factor. We apply the QL approach described above in our system as illustrated in Figure 1. As described previously, since the state represents the selected action, namely, the transmission power level, the Q-table is denoted by $Q^{\left(t\right)}\left(a^{\left(t\right)}\right)$, which indicates the preference transmission power of each mSBS in the transmission power level $a^{\left(t\right)}$ at iteration *t*. The new Q-value, i.e., $Q^{\left(t+1\right)}\left(a^{\left(t\right)}\right)$, is updated based on the previous Q-value and the current reward obtained from Equation (7). It can be represented as
(9)$$\begin{array}{c} Q^{\left(t+1\right)}\left(a^{\left(t\right)}\right)=\left(1-\alpha \right)Q^{\left(t\right)}\left(a^{\left(t\right)}\right)+\alpha \left\{R^{\left(t\right)}-Q^{\left(t\right)}\left(a^{\left(t+1\right)}\right)\right\}. \end{array}$$

**Optimal policy ($\pi^{*}$):** The transmission power control problem, which aims to find the optimal transmission power that maximizes the reward *R*, is formulated as follows:(10)$$\begin{array}{cc} \mathrm{Problem}\;:\; & \max_{a^{\left(t\right)}}R^{\left(t\right)}\\ \mathrm{Subject}\;\mathrm{to}\;:\; & a^{\left(t\right)}\in \left\{p_{1},p_{2},\ldots ,p_{L}\right\}. \end{array}$$

For this optimization problem, each agent exploits $Q(a^{\left(t\right)})$, which represents the expected cumulative sum of rewards, as follows:(11)$$\begin{array}{c} Q\left(a^{\left(t\right)}\right)=E[\sum_{t^{\prime }=t}^{\infty }\gamma R^{\left(t^{\prime }\right)}|a^{\left(t\right)},\pi ]. \end{array}$$

As mentioned earlier, each mSBS can obtain an optimal policy $\pi^{*}$ by choosing an action based on a decayed-epsilon greedy policy. Algorithm 1 presents the detailed procedure of the proposed multi-agent QL algorithm for UDmN. At every iteration, mSBS *i* in state $s^{\left(t\right)}$ chooses action $a^{\left(t\right)}$ based on a decayed-epsilon greedy policy. Then, mSBS *i* calculates the reward, and the Q-tables are updated.
**Algorithm 1** Multi-agent QL algorithm.1.Initialize $Q^{\left(t\right)}\left(a^{\left(t\right)}\right)$ of each mSBS.2.for Every iteration do3.  for
i=1:N
do4.    mSBS *i* at state $s^{\left(t\right)}$ choose an action $a^{\left(t\right)}$ based on a decayed-epsilon greedy policy.5.    
$$\begin{array}{c} a^{\left(t\right)}=\left\{\begin{array}{cc} \arg\max_{a^{\left(t\right)}\in A}Q^{\left(t\right)}\left(a^{\left(t\right)}\right), & \mathrm{with}\;1-\epsilon (e_{i})\\ \mathrm{random}\;\mathrm{action}, & \mathrm{with}\;\epsilon (e_{i}). \end{array}\right. \end{array}$$6.    Calculate the reward, $R^{\left(t\right)}$7.    Based on the reward, update Q-table as Equation (9)8.  end9.end

## 5. Numerical Examples

For the performance evaluation of the proposed system, we build a simulation program using MATLAB R2020a. In the network topology, the number of mSBSs that can conduct cooperative transmission is set to 4, and the number of interfering mSBSs is set to 15. The distance between the user and the mSBSs conducting cooperative transmission, $r_{c}$, is 200 m, while the distance between the user and the interfering mSBSs, $r_{i}$, is 500 m, as shown in Figure 2. In this model, we consider the rectangle model for the blockage effect model as [25]. For the channel model, we consider that each mSBS has an independent and identically distributed Nakagami fading channel and *m* = 3. Moreover, ς = 2, $\sigma^{2}$ = −174 dBm/Hz + 10log10($B_{s}$ + 10 dB), $B_{s}$ = 1 GHz, and $\gamma_{th}$ = −1. For the hyperparameters, α = 0.1, γ = 0.95, $\epsilon_{0}$ = 0.99, and ξ = 50. The parameters chosen in our framework are listed in Table 1. Our system is also compared with the following three schemes: (1) Random Action (Random): mSBS randomly chooses an action in each iteration. (2) Reward-Optimal (Op): The optimal solution can be obtained by the exhaustive search algorithm, where all possible states are considered. (3) No cooperative transmission scheme (NoCooper): mSBSs do not conduct cooperative transmission.

Figure 3 shows the accumulated reward of multiple agents, i.e., mSBSs, for β = 0.9 and *N* = 8 in a multi-agent environment, where β denotes the importance weight between the outage and the power efficiency gain. To obtain the results, we perform 1000 episodes. For Op, it shows the results of finding the optimal transmission power by exhaustively considering all possible cases. This method theoretically provides the highest average reward. Random denotes the approach where the transmission power is selected randomly without considering Q-values. This method is expected to have a lower average reward and exhibit greater variability, indicating the inefficiency of the random policy in adapting to the environment. RL represents the results based on the reinforcement learning algorithm described earlier, where each agent adjusts its transmission power based on its own distributed Q-table. Initially, the RL algorithm may yield lower rewards, but with an increasing number of episodes, the average reward from the RL approaches the reward obtained from the Op. This pattern indicates that the agents are learning better power adjustment strategies through their interactions with the environment. The non-cooperation transmission scheme (i.e., Nocooper) shows a low reward due to the insufficient handling of outage.

Figure 4 shows the accumulated reward of mSBSs for β = 0.95. Similar to Figure 3, the results are obtained over 1000 episodes. In Figure 4, because the β value is higher, more weight is given to outage, thus reflecting its increased importance in the reward calculation. As a result, the accumulated reward in Figure 4 is relatively higher than that in Figure 3. This is because a higher β value emphasizes the more significant reduction of outage, ultimately leading to a higher overall reward. The overall trend in Figure 4 remains consistent with that presented in Figure 3.

Figure 5 shows the relationship between the number of cooperative mSBSs and two key performance metrics: reward (*R*) and power efficiency gain ($P_{g}$). As the number of cooperative mSBSs increases, there is a notable trend where decreased power is required for transmission due to enhanced cooperation among the mSBSs. Simultaneously, the energy efficiency of the system improves as the number of cooperative mSBSs increases. The increase in energy efficiency is a direct result of the reduced power consumption, which is achieved through cooperative behavior among the mSBSs. This trend demonstrates that by increasing the number of cooperative mSBSs, it is possible to achieve significant improvements in energy efficiency while maintaining or even enhancing the overall system reward.

Figure 6 represents the effect of β in the reward function. As β increases, the reward function places greater weight on the outage probability, while a decrease in β emphasizes the energy efficiency. When β approaches 0, the reward is determined mainly by energy efficiency, which is ideal for minimizing power consumption. As β approaches 1, the reward is more influenced by the outage probability, making it more suitable for scenarios where the service reliability is prioritized. For instance, when β=0.7, the reward reaches its lowest value, reflecting the greater impact of outage probability. As β increases up to 0.7, all values decrease, focusing on the outage probability. However, after this point, the reward begins to increase while the outage probability drops sharply. At β=0.5, the reward function shows a balanced trade-off between the outage probability and energy efficiency, resulting in a moderate reward.

Figure 7 shows the variation in reward values as the path loss exponent is adjusted to values of 2, 3, and 4, where the path loss exponent determines the rate of signal attenuation over distance. For example, higher exponent values result in faster signal decay, which can lead to improved SINR due to reduced interference. In our approach, power control is applied to maximize reward by optimizing SINR, thereby reducing outage probability and enhancing energy efficiency. From this figure, we can see that higher path loss exponent values lead to fewer episodes required for learning convergence and ultimately yield higher reward values.

Figure 8 illustrates the effects of varying key RL parameters on the reward. Specifically, each subfigure shows the impact of distinct parameter adjustments: Figure 8a the discount factor, Figure 8b the learning rate, and Figure 8c the exploration parameter. From this figure, we can see that changes in each parameter affect the number of iterations required for the reward to converge, with higher values of these parameters typically resulting in a longer convergence time. For instance, as the discount factor increases, reflecting a greater emphasis on future rewards, the number of iterations required for convergence increases as shown in Figure 8a. However, the ultimate converged reward value remains largely unchanged. Similarly, in Figure 8b, a higher learning rate accelerates the convergence process, while the final reward value is minimally affected. Lastly, adjustments to the exploration parameter, as depicted in Figure 8c, demonstrate that a higher exploration rate slightly increases the convergence time, yet the overall performance outcome remains consistent.

Figure 9 shows the accumulated reward of mSBSs considering blockage effects as shown in Figure 2. In this figure, Interfering1 and Interfering2 represent scenarios where one and two interfering signals pass through walls, respectively, each experiencing a 20 dB penetration loss due to the blockage effect [25]. In the case of Interfering2, the two interfering signals experience cumulative attenuation, which further reduces the interference levels, improves SINR, reduces outage, and consequently increases the reward value. Conversely, Cooperative1 and Cooperative2 involve cooperative signals passing through one and two walls, respectively, with each signal experiencing a 20 dB loss. In Cooperative2, both cooperative signals are attenuated by 20 dB, weakening the cooperative signal strength. This reduction in SINR increases the outage, resulting in a decrease in the reward value. From these results, we can see that the QL-based power control effectively adjusts the transmission power to balance the energy efficiency and QoS requirements by accounting for the blockage effect on both interfering and cooperative signals as they pass through walls.

Figure 10 shows the performance of an algorithm that determines whether to participate in a QL approach based on channel state information (CSI) (e.g., received signal strength indicator, or RSSI), considering blockage effects due to walls [26]. The performance metrics evaluated are reward, outage probability, and energy efficiency. In this environment, we assume that the network consists of eight cooperative mSBSs, of which two are affected by wall blockage. In the Perfect CSI scenario, wall effects are accurately identified, allowing cooperative mSBSs affected by the wall to opt out of power control, thereby improving overall energy efficiency. Conversely, Imperfect CSI fails to distinguish wall penetration accurately, leading all signals to participate in power control, which increases the total transmission power. However, due to the significant signal attenuation by the wall, the outage probability remains unaffected in both Perfect and Imperfect CSI cases. Consequently, the reward value is higher under Perfect CSI conditions.

## 6. Conclusions

In this paper, we have presented a novel approach to enhancing energy efficiency in ultra-dense millimeter-wave network (UDmN) environments by leveraging coordinated multi-point transmission with joint transmission (CoMP-JT). Our approach integrates a multi-agent Q-learning-based power control scheme to optimize the trade-off between energy consumption and user outage probability. The key findings of our study can be summarized as follows: (1) Energy-efficient cooperative transmission: We demonstrated that cooperative transmission among multiple small base stations (mSBSs) using CoMP-JT can significantly reduce the power required for transmission while maintaining high signal-to-interference-noise ratio (SINR) for users at cell edges. (2) Multi-agent Q-learning-based power control: Utilizing a multi-agent tabular Q-learning (QL) approach, the proposed scheme allows each mSBS to adjust its transmission power in a distributed manner, while considering both energy consumption and user outage. (3) Simulation results: Through extensive simulations, we showed that our proposed scheme achieves optimal transmission power control, resulting in significantly improved network energy efficiency compared to conventional algorithms, such as those without transmit power control or with random control. In particular, our approach demonstrates higher accumulated rewards, reflecting better power efficiency and reduced user outages. Additionally, we validated the performance of an algorithm that determines whether each mSBS participates in power control based on channel state information (CSI), achieving further improvements in network efficiency. For future work, we plan to investigate a Q-learning approach that considers beam misalignment to mitigate blockage effects, aiming to further enhance energy efficiency in UDmNs.

## Figures and Tables

**Figure 1 sensors-24-07750-f001:**
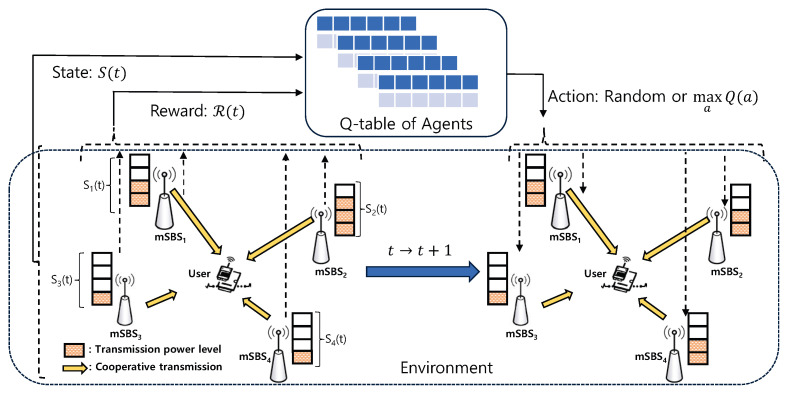
System model for a multi-agent Q-learning-based power control scheme in UDmN.

**Figure 2 sensors-24-07750-f002:**
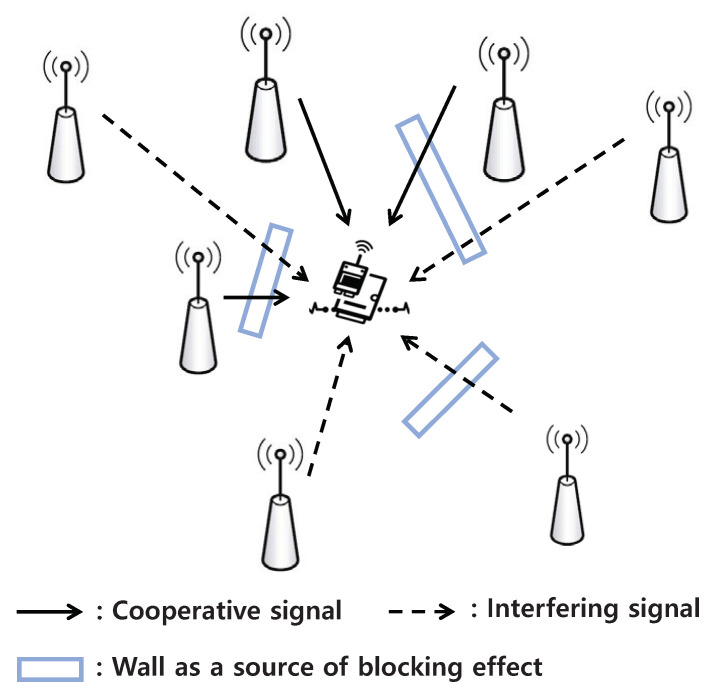
System topology.

**Figure 3 sensors-24-07750-f003:**
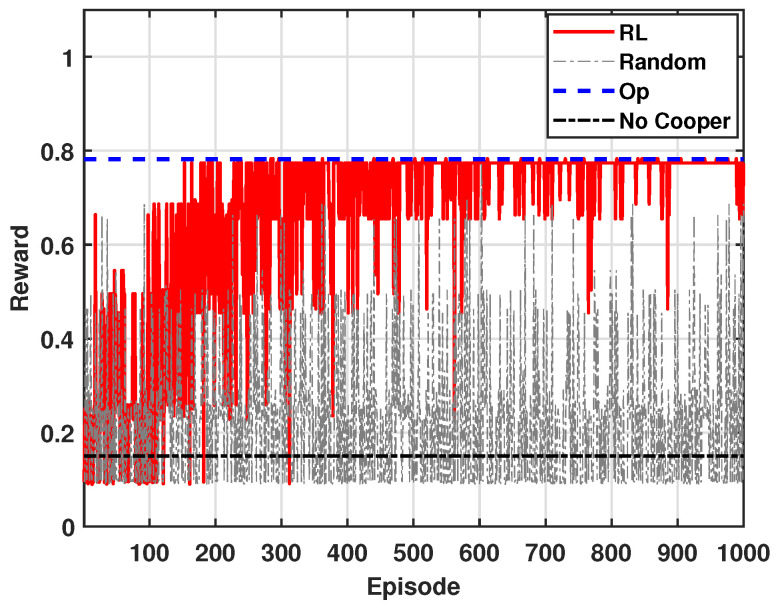
Accumulated average reward (β = 0.9).

**Figure 4 sensors-24-07750-f004:**
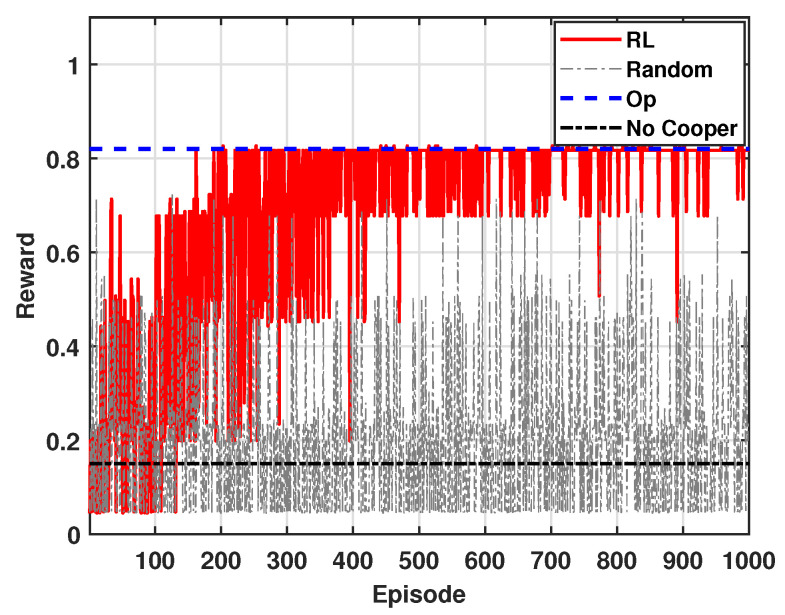
Accumulated average reward (β = 0.95).

**Figure 5 sensors-24-07750-f005:**
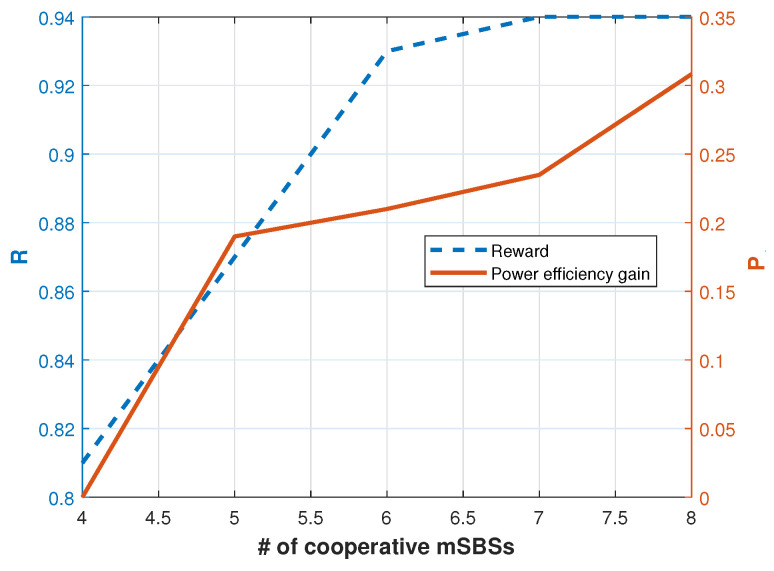
Accumulated average reward and power efficiency gain versus *N*.

**Figure 6 sensors-24-07750-f006:**
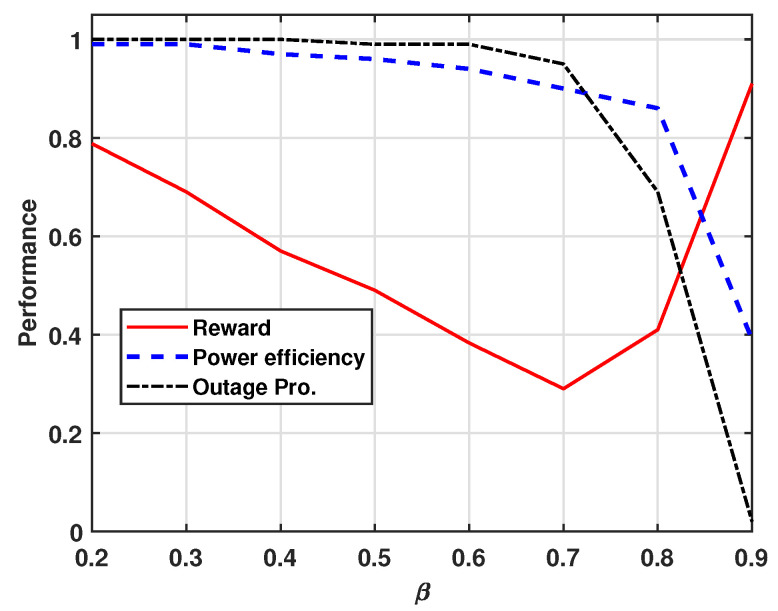
Performance of reward, power efficiency gain, and outage probability versus β.

**Figure 7 sensors-24-07750-f007:**
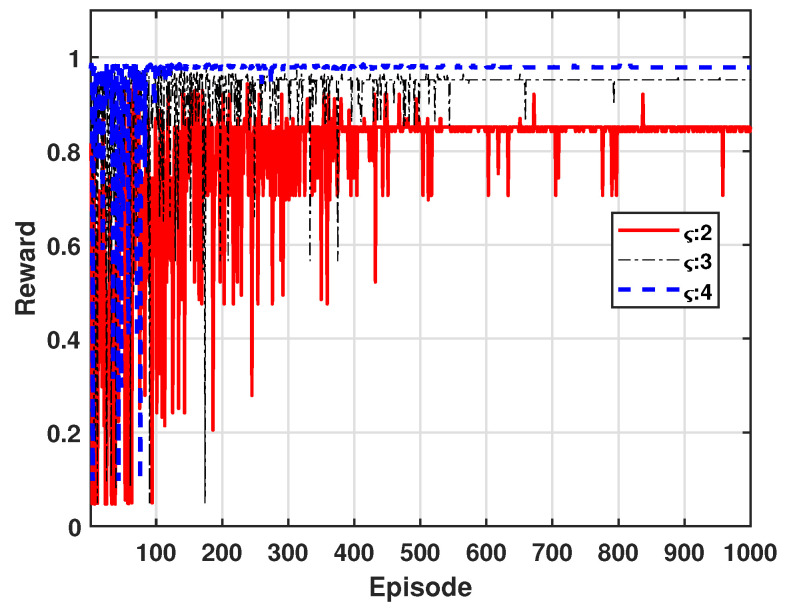
Accumulated average reward for path loss exponents.

**Figure 8 sensors-24-07750-f008:**
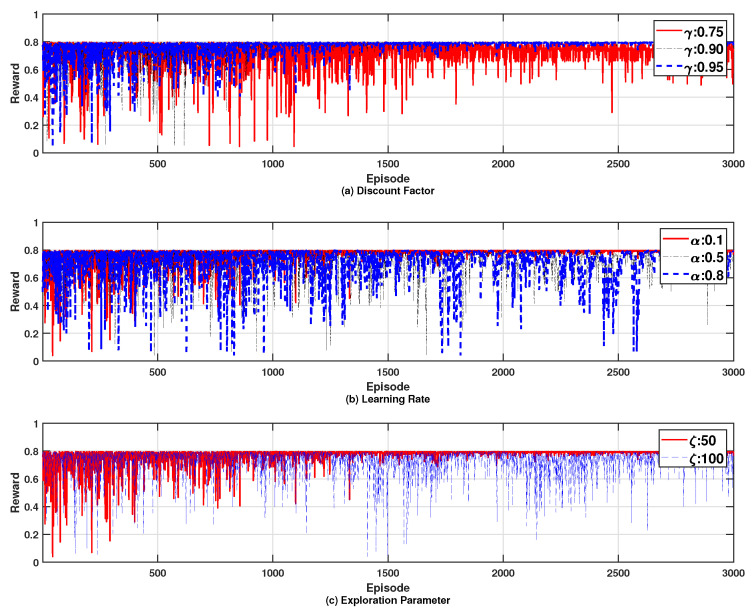
Effects of varying key RL parameters on the reward; (**a**) discount factor, (**b**) learning rate, and (**c**) exploration parameter.

**Figure 9 sensors-24-07750-f009:**
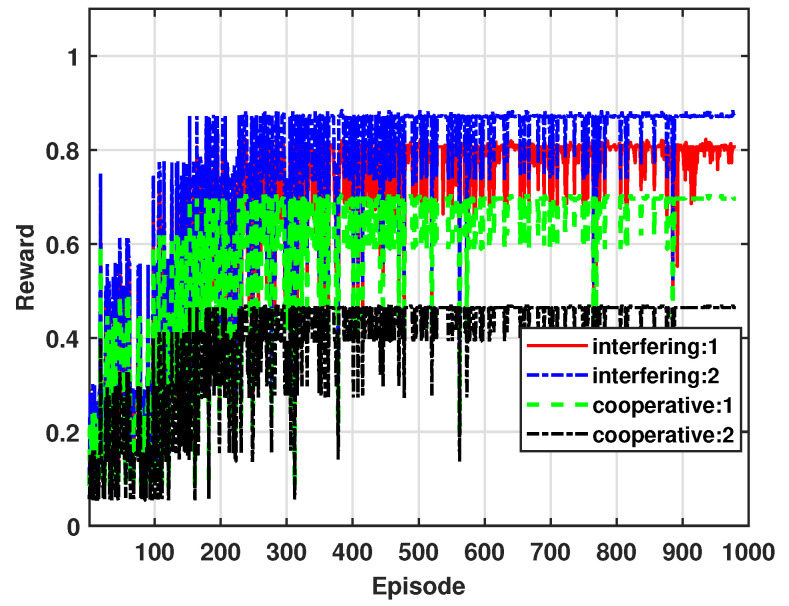
Accumulated average reward for blockage effects.

**Figure 10 sensors-24-07750-f010:**
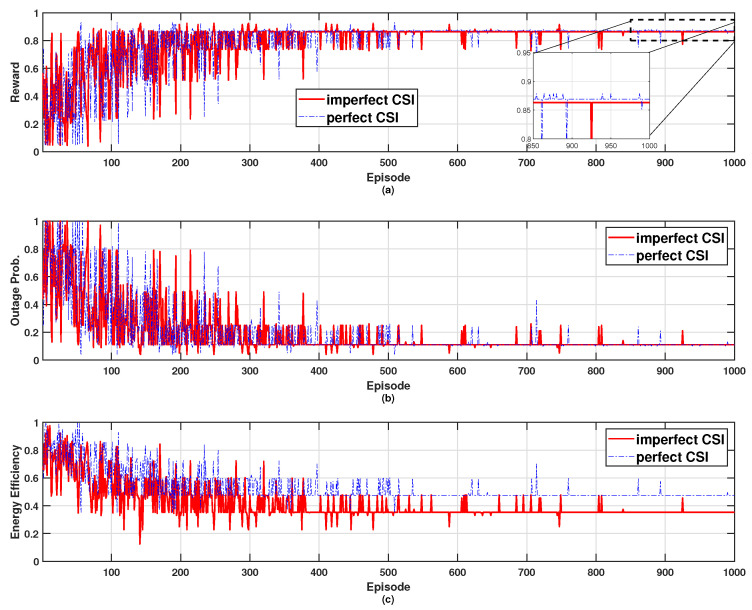
Effects of channel state information on the performance; (**a**) reward, (**b**) outage probability, and (**c**) energy efficiency.

**Table 1 sensors-24-07750-t001:** Simulation and hyperparameters.

Parameter	Value
$\sigma^{2}$	−174 dBm/Hz + 10log_10_($B_{s}$ + 10 dB)
$B_{s}$	1 GHz
ς	2
$r_{c}$	200 m
$r_{i}$	500 m
*m*	3
$\gamma_{th}$	−1
$p_{L}$	30 dBm
*L*	5
α	0.1
β	0.9, 0.95
γ	0.95
$\epsilon_{0}$	0.99
ξ	50

## Data Availability

Data are contained within the article.

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
