# Peer review of "Energy-Efficient Cooperative Transmission in Ultra-Dense Millimeter-Wave Network: Multi-Agent Q-Learning Approach"

_sensors, 2024, doi:10.3390/s24237750_

Round 1
Reviewer 1 Report
Comments and Suggestions for Authors
The manuscript entitled 'Energy Efficient Cooperative Transmission in Ultra-Dense Millimeter-Wave Network: Multi-Agent Q-learning Approach' presents a novel approach combining CoMP-JT with multi-agent Q-learning for power control in UDmN. While the use of multi-agent Q-learning for power control is not entirely new, the application to cooperative transmission in mmWave networks and the specific reward function design considering both energy efficiency and outage probability is innovative. The manuscript is generally well-structured and follows a logical flow. However, I have several concerns that must be addressed before reconsideration.
1) The comparison with existing solutions is somewhat limited, particularly regarding other machine learning approaches in similar scenarios.
2) The Q-learning framework is appropriately formulated for the distributed power control problem. However, the convergence analysis of the multi-agent Q-learning algorithm is missing. And the impact of imperfect channel state information is not considered in the theoretical model. The author should add theoretical analysis of the algorithm's convergence properties and include the effects of channel estimation errors and delayed feedback in the system model.
3) There should be a conduct sensitivity analysis for key parameters like learning rate, discount factor, and exploration rate.
4) The SINR model presented in Section 3.1 makes some simplifying assumptions that may limit its practical applicability. While the Nakagami fading model is appropriate for mmWave channels, the author does not adequately address the impact of beam misalignment and blockage effects, which are crucial in mmWave systems. He/She should consider incorporating these practical impairments into their system model.
5) The reward function design (Equation 7) combines outage probability and power efficiency using a weighted sum approach. While this is reasonable, the author does not provide sufficient justification for the choice of the weighting parameter β. A more rigorous analysis of how different β values affect the trade-off between reliability and energy efficiency would strengthen the technical contribution.
6) The simulation setup could benefit from more realistic assumptions. The fixed geometric placement of BSs (200m for cooperative BSs and 500m for interfering BSs) is overly simplistic. Real urban deployments would have more varied distances and irregular patterns. Additionally, the choice of simulation parameters (particularly the path loss exponent =2) seems optimistic for mmWave frequencies.
7) References [1]-[4] provide good background on mmWave networks and ultra-dense deployments. However, the coverage of multi-agent reinforcement learning literature is somewhat limited. Key papers on multi-agent Q-learning in wireless networks are missing.
There are occasional grammatical errors, particularly in subject-verb agreement.
Inconsistent use of articles (a/an/the) throughout the text.
Reviewer 2 Report
Comments and Suggestions for Authors
Dear Author,
Attached is the review file.
Best regards,
